# BANANAS: Bayesian Optimization with Neural Networks for Neural Architecture Search

## Abstract

Neural Architecture Search (NAS) has seen an explosion of research in the past few years. A variety of methods have been proposed to perform NAS, including reinforcement learning, Bayesian optimization with a Gaussian process model, evolutionary search, and gradient descent. In this work, we design a NAS algorithm that performs Bayesian optimization using a neural network model.

We develop a path-based encoding scheme to featurize the neural architectures that are used to train the neural network model. This strategy is particularly effective for encoding architectures in cell-based search spaces. After training on just 200 random neural architectures, we are able to predict the validation accuracy of a new architecture to within one percent of its true accuracy on average, for popular search spaces. This may be of independent interest beyond Bayesian neural architecture search.

We test our algorithm on the NASBench (Ying et al. 2019) and DARTS (Liu et al. 2018) search spaces, and we show that our algorithm outperforms other NAS methods including evolutionary search, reinforcement learning, AlphaX (Wang et al. 2019), ASHA, DARTS, and TPE. Our algorithm is over 100x more efficient than random search, and 3.8x more efficient than the next-best algorithm on the NASBench dataset. As there have been problems with fair and reproducible experimental evauations in the field of NAS, we adhere to the recent NAS research checklist (Lindauer and Hutter 2019) to facilitate NAS research. In particular, our implementation has been made publicly available, including all details needed to fully reproduce our results.

## 1 Introduction

Since the deep learning revolution in 2012, neural networks have been growing increasingly more specialized and more complex (Krizhevsky et al., 2012; Huang et al., 2017; Szegedy et al., 2017). Developing new state-of-the-art architectures often takes a vast amount of engineering and domain knowledge. A new area of research, neural architecture search (NAS), seeks to automate this process. Since the popular NASNet paper (Zoph & Le, 2017), there has been a flurry of research on neural architecture search (Liu et al., 2018a; Pham et al., 2018; Liu et al., 2018b; Kandasamy et al., 2018; Elsken et al., 2018; Jin et al., 2018; Sciuto et al., 2019; Cortes et al., 2017; Baker et al., 2016; Liu et al., 2017). Many methods have been proposed for NAS, including random search, evolutionary search, reinforcement learning, Bayesian optimization, and gradient descent. Each technique has its advantages and disadvantages. For example, reinforcement learning with a controller neural network can learn very complex policies. However, as some past work has argued, NAS is an optimization problem – find the architecture with the lowest validation error – so there is no explicit need to maintain a state and transition function, and this approach may require evaluating a larger number of architectues to achieve the same result as other methods (Kandasamy et al., 2018; Elsken et al., 2018; Jiang et al., 2017). Alternatively, Bayesian optimization with a Gaussian process model is a leading method for zeroth order optimization of expensive-to-evaluate functions, which is precisely the setting of deep learning model selection. Although Bayesian optimization has seen great success in hyperparameter optimization (Golovin et al., 2017; Li et al., 2016) for deep models, a host of difficulties arise when using Bayesian optimization for neural architecture search. For example, while there is a substantial body of work on Bayesian optimization over Euclidean spaces, using Bayesian optimization for NAS has so far required specifying a distance function between neural

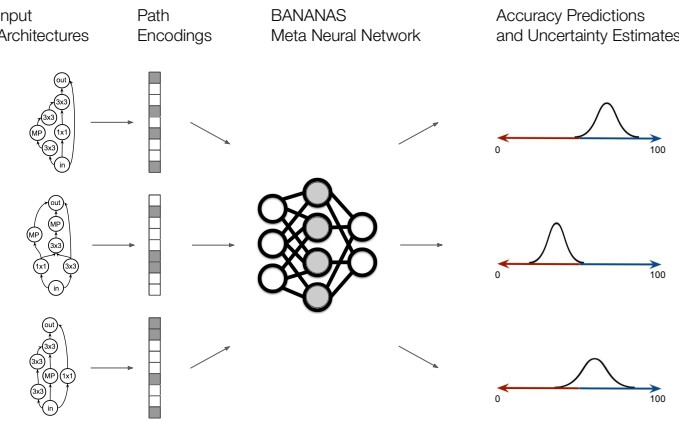

Figure 1: Illustration of the meta neural network in the BANANAS algorithm.

architectures, in order to define a surrogate model. This is often a cumbersome task involving tuning hyperparameters of the distance function (Kandasamy et al., 2018; Jin et al., 2018). Furthermore, Bayesian optimization most commonly uses a Gaussian process (GP) model, and it can be quite challenging to achieve highly accurate prediction performance with GPs, given the potentially high dimensional input architectures (Elsken et al., 2018).

In this work, we run Bayesian optimization with a neural network model. That is, in every iteration of Bayesian optimization, we train a meta neural network to predict the accuracy of unseen neural architectures in the search space. This procedure avoids the aforementioned problems with Bayesian optimization NAS: the model is powerful enough to predict neural network accuracies, and there is no need to construct a distance function between neural networks by hand. Furthermore, using a neural network model scales far better than a GP model, as it avoids the computationally intensive matrix inversion operation.

Training a meta neural network to predict the accuracy of neural architectures is a challenging task. The majority of popular NAS algorithms are deployed over a directed acyclic graph (DAG) search space – the set of possible architectures is comprised of the set of all DAGs of a certain size, together with all possible combinations of operations on each node (Zoph & Le, 2017; Pham et al., 2018; Liu et al., 2018b; Ying et al., 2019). This poses a roadblock when using a meta neural network to predict the accuracies of neural architectures, since graph structures are difficult for neural networks to understand directly from the adjacency matrix (Zhou et al., 2018). We use a *path-based encoding scheme* to encode a neural architecture, which drastically improves the predictive accuracy of our meta neural network. Each binary feature represents a unique path through the DAG from the input layer to the output layer. This method improves the accuracy of our meta neural network by a factor of 3. With just 200 training points, our meta neural network is able to predict the accuracy of unseen neural networks to within one percent on average on the popular NASBench dataset (Ying et al., 2019). This may be of interest beyond Bayesian neural architecture search.

In our Bayesian optimization NAS algorithm, we train an ensemble of neural networks to predict the mean and variance of candidate neural architectures, from which we compute the upper confidence bound (UCB) acquisition function (Srinivas et al., 2009). Finally, we use a mutation algorithm to optimize the acquisition function. We compare our NAS algorithm against a host of black-box optimization NAS algorithms including regularized evolution, reinforcement learning, Bayesian optimization with a GP model, AlphaX (Wang et al., 2018), ASHA (Li & Talwalkar, 2019), DARTS (Liu et al., 2018b), and TPE (Bergstra et al., 2011). On the NASBench dataset, our method sees significant improvements compared to all other algorithms we tried across 200 trials. It takes the next-best algorithm 3.8 times the number of queries to achieve the same mean test accuracy as our method, and it takes random search more than 100 times the number of queries. On the search space from DARTS, when given a budget of 100 neural architecture queries for 50 epochs each, our algorithm achieves a best of 2.57% and average of 2.64% test error, which beats all NAS algorithms

with which we could fairly compare (e.g., same search space used, and same hyperparameters for the final training).

Our NAS algorithm has several moving parts, including the meta neural network, the path-based feature encoding of neural architectures, the acquisition function, and the acquisition optimization strategy. We run a thorough ablation study by removing each piece of the algorithm separately. We show all components are necessary to achieve the best performance. Finally, to adhere to recent calls for reproducible NAS research (Ying et al., 2019; Li & Talwalkar, 2019; Lindauer & Hutter, 2019), we address all points in the NAS research checklist (Lindauer & Hutter, 2019) and will make our anonymized code is available at `https://www.github.com/iclr474/bananas`.

**Our contributions.** We summarize our main contributions below.

- We propose a novel path-based encoding scheme for neural architectures. This is a method for featurizing neural architectures by enumerating paths along the DAG from the input node to the output node. When predicting the validation accuracy of neural networks, using this featurization drastically improves the performance compared to standard encodings such as the adjacency matrix encoding, making it feasible to use a meta neural network as a model in Bayesian optimization.
- We design a Bayesian optimization algorithm using a meta neural network model to perform NAS over cell-based search spaces. We use an ensemble of neural networks to output a mean and variance for each candidate architecture and then use the UCB acquisition function to guide the search. We perform an extensive study of our algorithm and show it performs on par or superior to several state-of-the-art NAS algorithms on two different search spaces.

## 2 RELATED WORK

Neural architecture search have been studied since at least the 1990s (Floreano et al., 2008; Kitano, 1990; Stanley & Miikkulainen, 2002), but the field was revitalized when NASNet gained significant attention (Zoph & Le, 2017). There has been a flurry of recent work, where the most popular techniques include evolutionary algorithms (Shah et al., 2018), reinforcement learning (Zoph & Le, 2017; Pham et al., 2018; Liu et al., 2018a; Tan & Le, 2019), Bayesian optimization (Kandasamy et al., 2018; Jin et al., 2018; Zhou et al., 2019), and gradient descent (Liu et al., 2018b). Recent papers have called for fair and reproducible experiments in the future (Li & Talwalkar, 2019; Sciuto et al., 2019; Ying et al., 2019). In this vein, the NASBench dataset was created, which contains over 400k neural architectures with precomputed training, validation, and test accuracy (Ying et al., 2019). A recent algorithm, AlphaX, uses a meta neural network to perform NAS (Wang et al., 2018), where the meta neural network is trained to make small changes to the current architecture, given adjacency matrix featurizations of neural architectures. See the recent survey on NAS (Elsken et al., 2018) for more information on NAS research. There are many additional recent NAS papers (Maziarz et al., 2018; Borsos et al., 2019; Wang et al., 2019).

Bayesian optimization is a leading technique for zeroth order optimization when function queries are expensive (Rasmussen, 2003; Frazier, 2018), and it has seen great success in hyperparameter optimization for deep learning (Rasmussen, 2003; Golovin et al., 2017; Li et al., 2016). Most work has focused on Euclidean or categorical input domains using a GP model (Rasmussen, 2003; Golovin et al., 2017; Frazier, 2018; Snoek et al., 2012). There has also been some previous work on using neural network models in Bayesian optimization (Snoek et al., 2015; Springenberg et al., 2016; Perrone et al., 2017), applied to the task of hyperparameter optimization.

There are several approaches for predicting the validation accuracy of neural networks using a layer-wise encoding of the neural architectures (Deng et al., 2017; Istrate et al., 2019), or predicting the learning curve of different neural architectures (Klein et al., 2017; Domhan et al., 2015; Baker et al., 2017). Ensembles of neural networks is a popular approach for uncertainty estimates (Beluch et al., 2018; Lakshminarayanan et al., 2017; Choi et al., 2016). For more discussion on related work, see Appendix A.

## 3 PRELIMINARIES

We give background on three key ingredients of NAS algorithms, as well as on Bayesian optimization.

**Search space.**   Before deploying a NAS algorithm, we must define the space of neural networks that the algorithm can search through. Perhaps the most common type of search space for NAS is a *cell-based search space* (Zoph & Le, 2017; Pham et al., 2018; Liu et al., 2018b; Li & Talwalkar, 2019; Sciuto et al., 2019; Ying et al., 2019). A *cell* consists of a relatively small section of a neural network, usually 6-12 nodes forming a directed acyclic graph (DAG). A neural architecture is then built by repeatedly stacking one or two different cells on top of each other sequentially, possibly separated by specialized layers. The layout of cells and specialized layers is called a *hyper-architecture*, and this is fixed, while the NAS algorithm searches for the best cells. The search space over cells consists of all possible DAGs of a certain size, where each node can be one of several operations such as $1 \times 1$ convolution, $3 \times 3$ convolution, or $3 \times 3$ max pooling. It is also common to set a restriction on the number of total edges or the in-degree of each node (Ying et al., 2019; Liu et al., 2018b). In this work, we focus on NAS over convolutional cell-based search spaces.

**Search strategy.**   The search strategy is the optimization method that the algorithm uses to find the optimal or near-optimal neural architecture from the search space. There are many varied search strategies, such as Bayesian optimization, evolutionary search, reinforcement learning, and gradient descent. In Section 4, we introduce a novel search strategy based on Bayesian optimization with a neural network model using a path-based encoding.

**Evaluation method.**   Many types of NAS algorithms consist of an iterative framework in which the algorithm chooses a neural network to train, computes its validation error, and uses this result to guide the choice of neural network in the next iteration. The simplest instantiation of this approach is to train each neural network in a fixed way, i.e., the algorithm has black-box access to a function that trains a neural network for $x$ epochs and then returns the validation error (Pham et al., 2018; Liu et al., 2018a; Shah et al., 2018). Algorithms with black-box evaluation methods can be compared by returning the architecture with the lowest validation error after a certain number of queries to the black-box function. There are also multi-fidelity methods, for example, when a NAS algorithm chooses the number of training epochs in addition to the architecture. In this work, we show our algorithm far outperforms several other NAS algorithms that use black box optimization, and we discuss multi-fidelity approaches in Section 6.

**Bayesian optimization.**   In applications of Bayesian optimization for deep learning, the typical goal is to find a neural architecture and/or set of hyperparameters that lead to an optimal validation error. Formally, Bayesian optimization seeks to compute $a^* = \arg\min_{a \in A} f(a)$, where $A$ is the search space, and $f(a)$ denotes the validation error of architecture $a$ after training on a fixed dataset for a fixed number of epochs. In the standard Bayesian optimization setting, over a sequence of iterations, the results from all previous iterations are used to model the topology of $f(A)$ using the posterior distribution of the model (often a Gaussian process). The next architecture is then chosen by optimizing an acquisition function such as expected improvement (EI) (Močkus, 1975) or upper confidence bound (UCB) (Srinivas et al., 2009). These functions balance exploration with exploitation during the iterative search. We discuss UCB in the next section, and we discuss other acquisition functions in Appendix B.

## 4 METHODOLOGY

In this section, we discuss all of the components of our NAS algorithm. Our Bayesian optimization method uses a meta neural network to predict the validation error of neural architectures from the search space. First we describe the featurization of neural architectures used in our algorithm. Then we describe how to predict mean and uncertainty estimates using an ensemble of meta neural networks. Finally, we describe the acquisition function and acquisition optimization strategy we use, and show how these pieces come together to form our final NAS algorithm.

**Architecture featurization via path-based encoding.**   Prior work aiming to encode or featurize neural networks have proposed using an adjacency matrix-based approach (Wang et al., 2018; Ying et al., 2019; Deng et al., 2017; Baker et al., 2017).

The adjacency matrix encoding gives an arbitrary ordering to the nodes, and then gives a binary feature for an edge between node i and node j, for all i, j. Then a list of the operations at each node

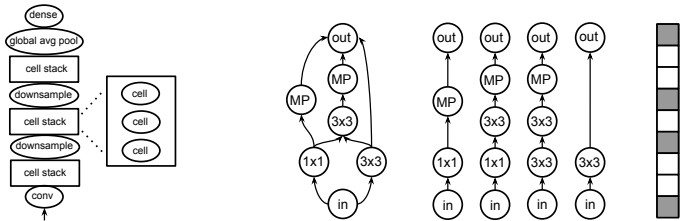

Figure 2: Hyper-architecture of a NASBench neural network (left). Example of a path encoding of a cell (right).

must also be included in the encoding. This is a challenging data structure for a NAS algorithm to interpret, because it relies on an arbitrary indexing of the nodes, and each feature is highly dependent on one another. For example, an edge from the input to node 2 is useless if there is no path from node 2 to the output. Also, even if there is an edge from node 2 to the output, it matters a lot whether node 2 is a CONV or a POOL operation. Ying et al. tested another encoding that is very similar to the adjacency matrix encoding, but the features of each edge are continuous.

We introduce a path-based encoding and show that it substantially increases the performance of our deep predictive uncertainty model. Each feature of the path encoding corresponds with a directed path from the input to the output of an architecture cell (for example: input→CONV 1x1→CONV 3x3→MP→output). To encode an architecture, we simply check which of the possible paths it contains (i.e. which paths are present in the cell), and write this as a binary vector. Therefore, we do not need to arrange the nodes in any particular order. Furthermore, the features are not nearly as dependent on one another as they are in the adjacency matrix encoding.

The total number of paths is $\sum_{i=0}^{n} q^i$ where $n$ denotes the number of nodes in the cell, and $q$ denotes the number of operations for each node. For example, the search space from NASBench (Ying et al., 2019) has $\sum_{i=0}^{5} 3^i = 364$ possible paths, and the search space from DARTS (Liu et al., 2018b) has $4 \cdot \sum_{i=0}^{4} 8^i = 18724$ possible paths. See Figure 2 (right). Our featurization performs very well on the most popular search spaces, however it may not scale to large search spaces. We discuss this further in Section 6.

**Acquisition function and optimization.**   Many acquisition functions used in Bayesian optimization can be approximately computed using a mean and uncertainty estimate for each input datapoint. We train an ensemble of five meta neural networks in order to predict mean and uncertainty estimates for new neural architectures. Concretely, we predict the validation error, as well as confidence intervals around the predicted validation error, of neural architectures that we have not yet observed. Suppose we have an ensemble of $M$ neural networks, $\{\hat{f}_m\}_{m=1}^{M}$, where each $\hat{f}_m : A \to \mathbb{R}$. We compute an estimate of the UCB acquisition function via

$$\phi_{\text{UCB}}(a) = \text{UCB}\left(\{\hat{f}_m(a)\}_{m=1}^{M}\right) = \hat{f} - \beta\hat{\sigma}, \tag{1}$$

where parameters $\hat{f} = \frac{1}{M}\sum_{m=1}^{M} \hat{f}_m(a)$, and $\hat{\sigma} = \sqrt{\frac{\sum_{m=1}^{M}(\hat{f}_m(a)-\hat{f})^2}{M-1}}$. Here we use UCB to denote an estimate of a lower confidence bound (since we are performing minimization) for the predictive uncertainty, and $\beta$ is a tradeoff parameter to control exploration vs exploitation.

In each iteration of Bayesian optimization, our goal is to find the neural network from the search space which minimizes the acquisition function. Evaluating the acquisition function for every neural network in the search space is computationally infeasible. Instead, we optimize the acquisition function via a mutation procedure, where we randomly mutate the best architectures that we have trained so far, many times, and then select the architecture from this set which minimizes the acquisition function. A cell is mutated by either adding an edge, removing an edge, or changing one of the operations with some probability.

**BANANAS: Bayesian optimization with neural architectures for NAS.**   Now we present our full NAS algorithm. At the start, we draw $t_0$ random architectures from the search space. Then

we start an iterative process, where in iteration $t$, we train an ensemble of meta neural networks on architectures $a_0, \ldots, a_{t-1}$. Each meta neural network in the ensemble has a different random initialization of the weights and a different random ordering of the training set. We create a candidate set of architectures by mutating the best architectures we have seen so far, and we choose to train the candidate architecture which minimizes the UCB acquisition function. See Algorithm 1.

---

**Algorithm 1** BANANAS

---

**Input:** Search space $A$, dataset $D$, parameters $t_0,\ T,\ M,\ c,\ k$, acquisition function $\phi$

Draw $t_0$ architectures $a_0, \ldots, a_{t_0}$ uniformly at random from $A$ and train them on $D$

Denote $f(a)$ as the validation error of $a$ after training

For $t$ from $t_0$ to $T$,

1. Train an ensemble of meta neural networks on $\{(a_0, f(a_0)), \ldots, (a_t, f(a_t))\}$
2. Generate a set of $c$ candidate neural architectures from $A$ by randomly mutating the $k$ architectures $a$ from $\{a_0, \ldots, a_t\}$ with the lowest value of $f(a)$
3. For each candidate architecture $a$, evaluate the acquisition function $\phi(a)$
4. Denote $a_{t+1}$ as the candidate architecture with minimum $\phi(a)$, and evaluate $f(a_{t+1})$

**Output:** $a^* = \mathrm{argmin}_{t=0,\ldots,T} f(a_t)$

---

## 5 EXPERIMENTS

In this section, we discuss our experimental setup and results. We give experimental results on the performance of the meta neural network itself, as well as the full NAS algorithm compared to several NAS algorithms. We start by explaining the two search spaces/datasets that we used.

**Search space from NASBench.** The NASBench dataset is a tabular dataset designed to facilitate NAS research and fair comparisons between NAS algorithms (Ying et al., 2019). It consists of over 423,000 unique neural architectures from a cell-based search space, and each architecture comes with precomputed training, validation, and test accuracies for 108 epochs on CIFAR-10. Most NAS algorithms take several GPU days to complete one trial, however, the NASBench dataset makes it feasible to run hundreds of trials of several NAS algorithms.

We give a description of the NASBench search space. For the full details, see (Ying et al., 2019). The search space consists of a cell with 7 nodes. The first node is the input, and the last node is the output. The remaining five nodes can be either $1 \times 1$ convolution, $3 \times 3$ convolution, or $3 \times 3$ max pooling. The cell can take on any DAG structure from the input to the output with at most 9 edges. The NASBench search space was chosen to contain ResNet-like and Inception-like cells (He et al., 2016; Szegedy et al., 2016). The hyper-architecture consists of nine cells stacked sequentially, with each set of three cells separated by downsampling layers. The first layer before the first cell is a convolutional layer, and the hyper-architecture ends with a global average pooling layer and dense layer. See Figure 2 (left).

**Search space from DARTS.** One of the most popular convolutional cell-based search spaces is the one from DARTS (Liu et al., 2018b), used for CIFAR-10. The search space consists of two cells with 6 nodes each: a convolution cell and a reduction cell, and the hyper-architecture stacks the convolutional and reduction cells. For each cell, the first two nodes are the inputs from the previous two cells in the hyper-architecture. The next four nodes each contain exactly two edges as input, such that the cell forms a connected DAG. Each *edge* can take one of seven operations: $3 \times 3$ and $5 \times 5$ separable convolutions, $3 \times 3$ and $5 \times 5$ dilated separable convolutions, $3 \times 3$ max pooling, $3 \times 3$ average pooling, identity, and *zero* (this is in contrast to NASBench, where the *nodes* take on operations).

### 5.1 NASBENCH EXPERIMENTS

First we evaluate the performance of the meta neural network on the NASBench dataset. The meta neural network consists of a sequential fully-connected neural network. The number of layers is set to 10, and each layer has width 20. We use the Adam optimizer with a learning rate of $0.1$, and use a mean absolute error loss function. We train the meta neural network on 200 architectures, and test

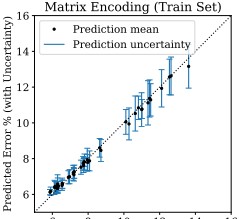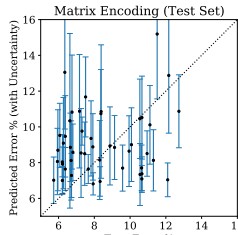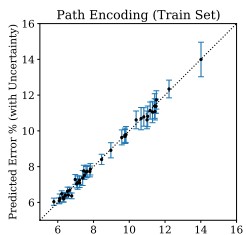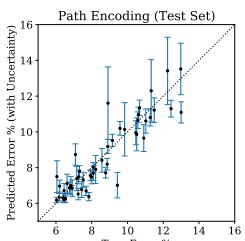

Figure 3: Predictive uncertainty estimates for architecture validation error under our ensemble model. We train this model on 200 architectures, and test the adjacency matrix encoding and path encoding on a held-out set of architectures as well as on a subset of the training set.

| Training set size | 20 | | 100 | | 500 | |
|---|---|---|---|---|---|---|
| | Train | Test | Train | Test | Train | Test |
| **Adjacency matrix encoding** | 0.507 | 2.977 | 0.575 | 2.626 | 0.633 | 2.618 |
| **Path-based encoding** | 0.384 | 2.335 | 0.477 | 1.178 | 0.317 | 0.644 |

Table 1: Meta neural network performance for different featurizations. We report the mean average percent error on the training set and the test set for three sizes of the training set.

the standard adjacency matrix encoding as well as the path encoding discussed in Section 4. See Figure 3. The path-based encoding outperforms the adjacency matrix encoding by a factor of 3 with respect to mean absolute error. In Appendix B, we plot the meta neural network performance with both featurizations for different training set sizes, and we report the training and test errors in Table 1.

Next, we compare BANANAS to five other algorithms. We give a brief description of each one.

**Random search.** The simplest baseline, random search, draws $n$ architectures at random and outputs the architecture with the lowest validation error. Despite its simplicity, multiple papers have concluded that random search is a competitive baseline for NAS algorithms (Li & Talwalkar, 2019; Sciuto et al., 2019).

**Regularized evolution.** This algorithm consists of drawing an initial set of cells at random, and then iteratively mutating the best architecture out of a sample of all architectures evaluated so far. The architectures with the worst validation error in each iteration are removed from the population. We used the same hyperparameters as in the NASBench paper, but changing the population size from 50 to 30 to account for fewer total queries.

**Reinforcement Learning.** We used the NASBench implementation of reinforcement learning for NAS (Ying et al., 2019) based on the REINFORCE algorithm (Williams, 1992). We used this algorithm because prior work has shown that a 1-layer LSTM controller trained with PPO is not effective on the NASBench dataset (Ying et al., 2019).

**AlphaX.** AlphaX casts NAS as a reinforcement learning problem, using a neural network to guide the search (Wang et al., 2018). Each iteration, a neural network is trained to select the best action, such as making a small change to, or growing, the current architecture.

**TPE.** Tree-structured Parzen estimator (TPE) is a hyperparameter optimization algorithm based on adaptive Parzen windows. We used the nasbench implementation (Ying et al., 2019).

**Bayesian optimization with a GP model.** We set up Bayesian optimization with a Gaussian process model and UCB acquisition. In the Gaussian process, we set the distance function between two neural networks as the sum of the Hamming distances between the adjacency matrices and the list of operations. Note we also used a path-based distance function in our ablation study.

**Experimental setup and results.** Each NAS algorithm is given a budget of 150 queries. That is, each algorithm can train and output the validation error of at most 150 architectures. Every 10 iterations, we return the architecture with the best validation error found by each algorithm so far.

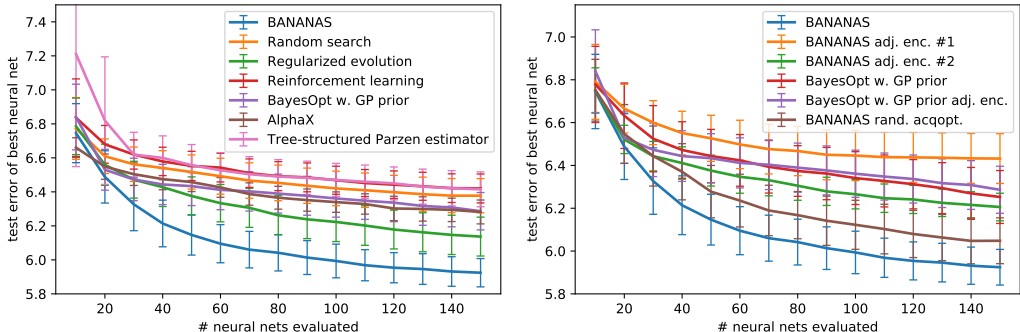

Figure 4: Performance of BANANAS compared to benchmarks (left). Ablation study of BANANAS (right). The error bars represent the 30% to 70% percentile range.

After all NAS algorithms have completed, we output the test error for each returned architecture. We ran 200 trials for each algorithm. For BANANAS, we use an ensemble of size 5 and UCB with $\beta = 0.5$ as in Equation 1. In each iteration, we chose the top ten candidate architectures to query, to improve runtime and so that the algorithm can later be parallelized. Although this strategy may not perform well for continuous search spaces, we found the NASBench search space was discrete enough that this method performs well.

We compared against other black-box optimization algorithms, so we report test error with respect to the number of queries. We note that a plot with respect to wall-clock time would look nearly identical, since all NAS algorithms finish the set of 150 queries in roughly the same amount of time. The fastest, random search, takes 46.5 TPU hours, while BANANAS, the slowest, takes 47.0 TPU hours (the average time to train a NASBench architecture is 18.6 minutes on a TPU, reported by (Ying et al., 2019)). See Figure 4 (left). BANANAS significantly outperforms all other baselines, and to the best of our knowledge, BANANAS achieves state-of-the-art error on NASBench in the 100-150 queries setting. The standard deviation among 200 trials for all NAS algorithms were between 0.16 and 0.22. BANANAS had the lowest standard deviation, and regularized evolution had the highest standard deviation.

**Ablation study.**   Our NAS algorithm has several moving parts, including the meta neural network model, the path-based feature encoding of neural architectures, the acquisition function, and the acquisition optimization strategy. We run a thorough ablation study by removing each piece of the algorithm separately. In particular, we compare against the following algorithms: *(1)* BANANAS where the featurization of each neural architecture is not the path-based encoding, but instead the common adjacency matrix and one-hot list of operations encoding; *(2)* BANANAS where the featurization of each neural architecture is the continuous variant of the adjacency matrix encoding from the nasbench paper (Ying et al., 2019); *(3)* BANANAS with a GP model instead of a neural network model, where the distance function in the GP is computed as the Hamming distance between the path-encoding vectors; *(4)* BANANAS with a GP model, where the distance function is computed as the sum of the Hamming distances between the adjacency matrices and the list of operations; *(5)* BANANAS where the acquisition funtion is optimized by drawing 1000 random architectures, instead of a mutation algorithm. We found that BANANAS significantly outperformed all of these variants. The path-encoding had the greatest effect on performance, while the acquisition optimization strategy had the smallest effect on performance. See Figure 4 (right).

In Appendix B, we present several more experiments. We give each NAS algorithm a budget of 500 queries instead of 150, and we show that BANANAS still outperforms all other algorithms. We discuss a different method to use the NASBench dataset (querying random vs mean validation error) and we present results in this setting, showing that our algorithm still performs the best. We also test three other common acquisition functions with BANANAS.

| NAS Algorithm | Source | Test error | | Queries | Runtime | Method |
|---|---|---|---|---|---|---|
| | | Avg | Best | | | |
| SNAS | (Xie et al., 2018) | 2.85 | | | 1.5 | Gradient based |
| ENAS | (Pham et al., 2018) | | 2.89 | | 0.5 | RL |
| Random search | (Liu et al., 2018b) | 3.29 | | | 4 | Random |
| DARTS | (Liu et al., 2018b) | 2.76 | | | 5 | Gradient-based |
| ASHA | (Li & Talwalkar, 2019) | 3.03 | 2.85 | 700 | 9 | Successive halving |
| Random search WS | (Li & Talwalkar, 2019) | 2.85 | 2.71 | 1000 | 9.7 | Random |
| DARTS | Ours | 2.68 | **2.57** | | 5 | Gradient-based |
| ASHA | Ours | 3.08 | 2.98 | 700 | 9 | Successive halving |
| BANANAS | Ours | **2.64** | **2.57** | 100 | 11.8 | Neural BayesOpt |

Table 2: Comparison of the mean test error of the best architectures returned by three NAS algorithms. The runtime is in total GPU-days on a Tesla V100. Note that ASHA queries use varying numbers of epochs.

## 5.2 DARTS SEARCH SPACE EXPERIMENTS.

We test the BANANAS algorithm on the search space from DARTS. We give BANANAS a budget of 100 queries, where each query involves training a neural network for 50 epochs and reporting its validation error. We parallelize the algorithm by choosing 10 neural networks to train in each iteration of Bayesian optimization. The total computation is 11.8 GPU days.

To ensure a fair comparison by controlling all hyperparameter settings and hardware (Lindauer & Hutter, 2019), we re-trained the architectures from papers which used the DARTS search space and reported the final architecture. We report the mean test error over five random seeds of the best architectures found by BANANAS, DARTS, and ASHA (Li & Talwalkar, 2019). We trained each architecture using the default hyperparameters from Li and Talwalkar (Li & Talwalkar, 2019) for 600 epochs.

The BANANAS architecture achieved an average of 2.64% accuracy, which is state-of-the-art for this search-space and final training parameter settings. See Table 2. We also compare to other NAS algorithms using the DARTS search space, however, the comparison is not perfect due to differences in the pytorch version of the final training code (which appears to change the final percent error by $\sim 0.1\%$). We cannot fairly compare our method to recent NAS algorithms which use a larger search space than DARTS, or which train the final architecture for significantly more than 600 epochs (Laube & Zell, 2019; Liang et al., 2019; Zhou et al., 2019). Our algorithm significantly beats ASHA, a multi-fidelity zeroth order NAS algorithm, and is on par with DARTS, a first-order method. We emphasize that since BANANAS is a zeroth order method, it allows for easy parallelism, integration with optimizing other hyperparameters, easy implementation, and optimization with respect to other non-differentiable objectives.

In the appendix, we run through the NAS research checklist (Lindauer & Hutter, 2019).

## 6 CONCLUSION AND FUTURE WORK

In this work, we propose a novel method for neural architecture search. Our method uses Bayesian optimization with a meta neural network model, and we encode the neural architectures using a path-based encoding scheme. This allows the meta neural network to accurately predict the validation error of new neural architectures. Our NAS algorithm significantly outperforms all other baselines on the NASBench dataset and outperforms state-of-the-art algorithms on the search space from DARTS.

One of the most interesting follow-up ideas is to develop a multi-fidelity version of BANANAS. For example, incorporating a successive-halving approach to BANANAS could result in a significant decrease in the runtime without substantially sacrificing accuracy. It would also be interesting to run our algorithm on other search spaces and datasets, such as Penn TreeBank and ImageNet. Finally, while we demonstrated that our path-based featurization performs very well for the most popular cell-based search spaces, our featurization would have trouble scaling to large cell-based search

spaces, as its dimensionality is exponential in the number of nodes per cell. It would be interesting to modify our approach so that it scales better to larger search spaces.

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

## A    RELATED WORK CONTINUED

**Neural architecture search.**    Evolutionary algorithms for neural architecture search have been studied since at least the 1990s (Floreano et al., 2008; Kitano, 1990; Stanley & Miikkulainen, 2002). This research area was revitalized when NASNet gained significant attention (Zoph & Le, 2017). NASNet is a reinforcement learning algorithm for NAS which achieves state-of-the-art results on CIFAR-10 and PTB; however, the algorithm requires 3000 GPU days to train. NASNet spurred a number of follow-up work, eventually pushing the required computation for NAS on CIFAR-10 and PTB to under 10 GPU days (Pham et al., 2018; Liu et al., 2018a;b; Ying et al., 2019; Kandasamy et al., 2018; Jin et al., 2018; Shah et al., 2018). Some of the most popular techniques for NAS include evolutionary algorithms (Shah et al., 2018), reinforcement learning (Zoph & Le, 2017; Pham et al., 2018; Liu et al., 2018a; Tan & Le, 2019), Bayesian optimization (Kandasamy et al., 2018; Jin et al., 2018), and gradient descent (Liu et al., 2018b).

Recent papers have called for fair and reproducible experiments in the future (Li & Talwalkar, 2019; Sciuto et al., 2019; Ying et al., 2019). In this vein, the NASBench dataset was created, which contains over 400k neural architectures with precomputed training, validation, and test accuracy (Ying et al., 2019).

A recent algorithm, AlphaX, uses a meta neural network to perform NAS (Wang et al., 2018). This is a similar high-level idea to our work but with many key differences. Their search is progressive, and each iteration makes a small change to the current neural network, rather than choosing a completely new neural network. Therefore, their meta neural network is trained to optimize the selection of new actions as well as to predict the performance of the new progressive network. Furthermore, they use an adjacency matrix featurization for the inputs to the meta neural network, which we find to perform worse than our path-based encoding.

**Bayesian optimization.**    Bayesian optimization is a leading technique for zeroth order optimization when function queries are expensive (Rasmussen, 2003; Frazier, 2018), and it has seen great success in hyperparameter optimization for deep learning (Rasmussen, 2003; Golovin et al., 2017; Li et al., 2016). The majority of Bayesian optimization literature has focused on Euclidean or categorical input domains, and has used a GP model (Rasmussen, 2003; Golovin et al., 2017; Frazier, 2018; Snoek et al., 2012). There has also been some previous work on using neural network models in Bayesian optimization (Snoek et al., 2015; Springenberg et al., 2016), applied to the task of hyperparameter optimization. Recently, there have been a couple works that use Bayesian optimization with GPs for NAS (Kandasamy et al., 2018; Jin et al., 2018).

**Predicting neural network accuracy.**    There are several approaches for predicting the validation accuracy of neural networks, such as a layer-wise encoding of neural networks with an LSTM algorithm (Deng et al., 2017), and a layer-wise encoding and dataset features to predict the accuracy for neural network + dataset pairs (Istrate et al., 2019). There is also work in predicting the learning curve of neural networks for hyperparameter optimization (Klein et al., 2017; Domhan et al., 2015) or NAS (Baker et al., 2017) using Bayesian techniques.

**Hyperparameter optimization using Bayesian optimization with neural networks.**    Snoek et al. showed that Bayesian optimization can scale by using a neural network to set up a kernel and a posterior distribution, instead of a Gaussian process (Snoek et al., 2012). Springenberg et al. give similar results using a stochastic gradient Hamiltonian Monte Carlo approach (Springenberg et al., 2016). Perrone et al. show how to apply warm start to Bayesian optimization with neural networks (Perrone et al., 2017).

## B    ADDITIONAL EXPERIMENTS

In this section, we present more experiments for the meta neural network and for BANANAS. We also discuss our experimental setup in more detail.

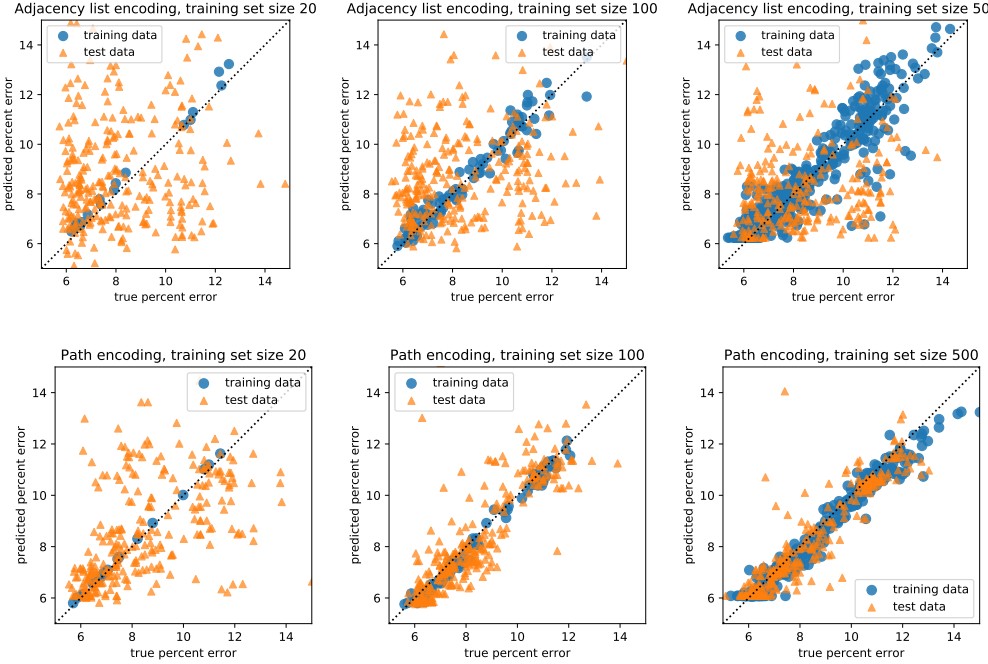

Figure 5: Performance of the meta neural network with adjacency matrix encoding (row 1) and path encoding (row 2). The training set is in blue, and the test set is in orange.

### B.1 META NEURAL NETWORK EXPERIMENTS

We plot the meta neural network performance with both featurizations for training sets of size 20, 100, and 500. We use the same experimental setup as described in Section 5. See Figure 5. We note that a training set of size 20 neural networks is realistic at the start of a NAS algorithm, and size 200 is realistic near the middle or end of a NAS algorithm.

### B.2 NASBENCH EXPERIMENTS

We run three more experiments on NASBench: running the algorithms with a budget of 500 queries rather than 150, testing out different acquisition functions with BANANAS, and using a slightly different validation error metric. All algorithms were run with 200 trials unless otherwise noted.

**NAS algorithms running for 500 queries.** In our main set of experiments, we gave each NAS algorithm a budget of 150 queries. That is, each NAS algorithm can only choose 150 architectures to train and evaluate. Now we give each NAS algorithm a budget of 500 queries. See Figure 6 (left). We largely see the same trends, although the gap between BANANAS and regularized evolution becomes smaller after query 400.

**Acquisition functions.** We tested BANANAS with three other common acquisition function in addition to UCB: expected improvement (EI) (Močkus, 1975), probability of improvement (PI) (Kushner, 1964), and Thompson sampling (TS) (Thompson, 1933). First we give the formal definitions of each acquisition function.

Suppose we have trained a collection of $M$ predictive models, $\{f_m\}_{m=1}^M$, where $f_m : A \to \mathbb{R}$. Following previous work (Neiswanger et al., 2019), we use the following acquisition function estimates for an input architecture $a \in A$:

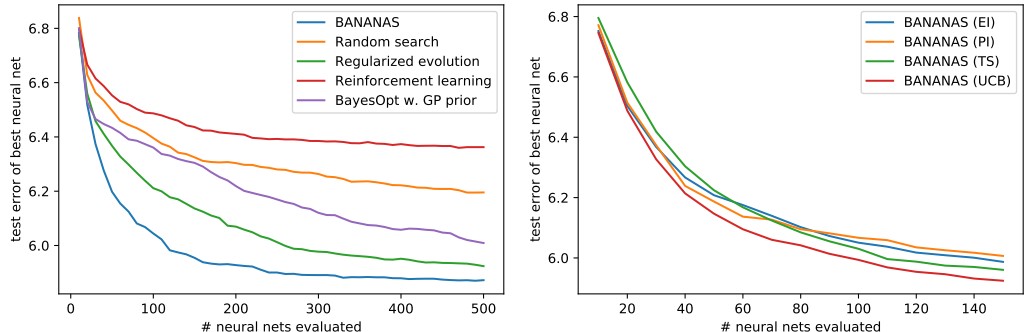

Figure 6: NAS algorithms with a budget of 500 queries (left). Comparison of different acquisition functions (right).

$$\phi_{\text{EI}}(a) = \mathbb{E}\left[\mathbb{1}\left[\hat{f}_m(a) > y_{\min}\right]\left(y_{\min} - \hat{f}_m(a)\right)\right] = \int_{-\infty}^{y_{\min}} (y_{\min} - y)\,\mathcal{N}\left(\hat{f}, \hat{\sigma}^2\right) dy \quad (2)$$

$$\phi_{\text{PI}}(x) = \mathbb{E}\left[\mathbb{1}\left[\hat{f}_m(x) > y_{\min}\right]\right] = \int_{-\infty}^{y_{\min}} \mathcal{N}\left(\hat{f}, \hat{\sigma}^2\right) dy \quad (3)$$

$$\phi_{\text{UCB}}(x) = \text{UCB}\left(\{\hat{f}_m(x)\}_{m=1}^M\right) = \hat{f} - \beta\hat{\sigma} \quad (4)$$

$$\phi_{\text{TS}}(x) = \hat{f}_{\tilde{m}}(x), \quad \tilde{m} \sim \text{Unif}\,(1, M) \quad (5)$$

In these acquisition function definitions, $\mathbb{1}(x) = 1$ if $x$ is true and 0 otherwise, and we are making a normal approximation for our model's posterior predictive density, where we estimate parameters $\hat{f} = \frac{1}{M}\sum_{m=1}^M \hat{f}_m(x)$, and $\hat{\sigma} = \sqrt{\frac{\sum_{m=1}^M (\hat{f}_m(x) - \hat{f})^2}{M-1}}$. In the UCB acquisition function, we use UCB to denote some estimate of a lower confidence bound for the posterior predictive density (note that we use a lower confidence bound rather than an upper confidence bound since we are performing minimization), and $\beta$ is a tradeoff parameter. In experiments we set $\beta = 0.5$.

See Figure 6 (right). We see that the acquisition function has little effect on the performance of BANANAS, though UCB performs best.

**Mean vs. random validation error.** In the NASBench dataset, each architecture was trained to 108 epochs three separate times with different random seeds. The NASBench paper conducted experiments by choosing a random validation error when querying each architecture, and then reporting the mean test error at the conclusion of the NAS algorithm. We found that the mismatch (choosing random validation error, but mean test error) added extra noise that would not be present during a real-life NAS experiment. Another way to conduct experiments on NASBench is to use mean validation error and mean test error. This is the method we used in Section 5 and the previous experiments from this section. Perhaps the most realistic experiment would be to use a random validation error and the test error corresponding to that validation error, however, the NASBench dataset does not explicitly give this functionality. In Figure 7, we give results in the setting of the NASBench paper, using random validation error and mean test error. We ran each algorithm for 500 trials. Note that the test error across the board is higher because the correlation between validation and test error is lower (i.e., validation error is noisier than normal). BANANAS still performs the best out of all algorithms that we tried. We see the trends are largely unchanged, although Bayesian optimization with a GP model performs better than regularized evolution. A possible explanation is that Bayesian optimization with a GP model is better-suited to handle noisy data than regularized evolution.

## C  BEST PRACTICES CHECKLIST FOR NAS RESEARCH

The area of NAS has seen problems with reproducibility, as well as fair empirical comparisons, even more so than most other areas of machine learning. Following calls for fair and reproducible NAS

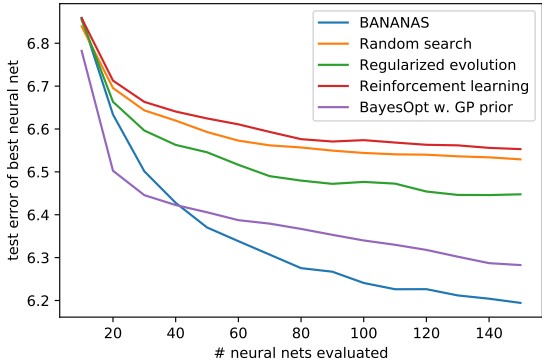

Figure 7: NAS experiments with random validation error and mean test error.

research (Li & Talwalkar, 2019; Sciuto et al., 2019; Ying et al., 2019), a best practices checklist was recently created (Lindauer & Hutter, 2019). In order to promote fair and reproducible NAS research, we address all points on the checklist, and we encourage future papers to do the same.

- *Code for the training pipeline used to evaluate the final architectures.* We used two of the most popular search spaces in NAS research, the NASBench search space, and the DARTS search space. For NASBench, the accuracy of all architectures were precomputed. For the DARTS search space, we released our fork of the DARTS repo, which is forked from the DARTS repo designed specifically for reproducible experiments (Li & Talwalkar, 2019), making trivial changes to account for pytorch 1.2.0.
- *Code for the search space.* We used the popular and publicly avaliable NASBench and DARTS search spaces with no changes.
- *Hyperparameters used for the final evaluation pipeline, as well as random seeds.* We left all hyperparameters unchanged. We trained the architectures found by BANANAS, ASHA, and DARTS five times each, using random seeds 0, 1, 2, 3, 4.
- *For all NAS methods you compare, did you use exactly the same NAS benchmark, including the same dataset, search space, and code for training the architectures and hyperparameters for that code?* Yes, we did this by virtue of the NASBench dataset. For the DARTS experiments, we used the reported architectures (found using the same search space and dataset as our method), and then we trained the final architectures using the same code, including hyperparameters. We compared different NAS methods using exactly the same NAS benchmark.
- *Did you control for confounding factors?* Yes, we used the same setup for all of our NASBench experiments. For the DARTS search space, we compared our algorithm to two other algorithms using the same setup (pytorch version, CUDA version, etc). Across training over 5 seeds for each algorithm, we used different GPUs, which we found to have no greater effect than using a different random seed.
- *Did you run ablation studies?* Yes, we ran a thorough ablation study.
- *Did you use the same evaluation protocol for the methods being compared?* Yes, we used the same evaluation protocol for all methods and we tried multiple evaluation protocols.
- *Did you compare performance over time?* In our main NASBench plots, we compared performance to number of queries, since all of our comparisons were to black-box optimization algorithms. Note that the number of queries is almost perfectly correlated with runtime. We computed the run time and found all algorithms were between 46.5 and 47.0 TPU hours.
- *Did you compare to random search?* Yes.
- *Did you perform multiple runs of your experiments and report seeds?* We ran 200 trials of our NASBench experiments. Since we ran so many trials, we did not report random seeds. We ran four total trials of BANANAS on the DARTS search space. Currently we do not have a fully deterministic version of BANANAS on the DARTS search space (which would be harder to implement as the algorithm runs on 10 GPUs). However, the average final error across trials was within 0.1%.

- *Did you use tabular or surrogate benchmarks for in-depth evaluations* Yes, we used NAS-Bench.
- *Did you report how you tuned hyperparameters, and what time and resources this required?* We performed light hyperparameter tuning for the meta neural network to choose the number of layers, layer size, learning rate, and number of epochs. In general, we found our algorithm to work well without hyperparameter tuning.
- *Did you report the time for the entire end-to-end NAS method?* We reported time for the entire end-to-end NAS method.
- *Did you report all details of your experimental setup?* We reported all details of our experimental setup.

