# OpenReview forum: "BANANAS: Bayesian Optimization with Neural Networks for Neural Architecture Search"
_ICLR.cc/2020/Conference — Reject_

### Official Review · AnonReviewer3 · 2019-10-18
**Official Blind Review #3**

**Rating:** 3

**Review:**

The paper considers the neural architecture search using Bayesian optimisation. The paper first propose a path-based encoding scheme to featurise the neural architectures. A path encoding of a cell is created by enumerating all possible paths from the input node to the output node. Then, they propose to train 5 neural networks and ensemble these networks to get the prediction (including the predictive value and uncertainty). The paper optimises the acquisition function via a mutation procedure, where we randomly mutate the best architectures that we have trained so far, many times, and then select the architecture from this set which minimizes the acquisition function.

While the writing is readable and the experiments seem promising, the reviewer thinks that the novelty and contribution of the paper are limited. Particularly, using 5 neural networks for estimating the uncertainty is less convincing. This is because it would have been better if we can train a Bayesian neural network to provide the uncertainty quantification directly.


Minor:
Page 5: “randomly randomly”


**Experience Assessment:**

I have published in this field for several years.

**Review Assessment: Checking Correctness Of Derivations And Theory:**

I assessed the sensibility of the derivations and theory.

**Review Assessment: Checking Correctness Of Experiments:**

I assessed the sensibility of the experiments.

**Review Assessment: Thoroughness In Paper Reading:**

I read the paper at least twice and used my best judgement in assessing the paper.

---

> ### Author Response · Authors · 2019-11-06
> **Reply to review #3**
>
> Thank you for your comments. Your only specific complaint is that we did not use a Bayesian neural network, and we believe that this comment alone does not justify a poor rating. We agree that using a Bayesian neural network may be an interesting alternative strategy. However, our ensemble approach is well-justified as we explain below.
>
> There have been a number of works that show the benefits of the ensembling approach to predictive uncertainty over using a classic Bayesian neural network. For example, ensembles can be trained in parallel, use less memory, do not require approximate Bayesian inference algorithm (MCMC or variational inference) and in some cases have been shown empirically to provide better predictive uncertainty estimates [1], [2], [3]. Many of these papers have shown good performance using an ensemble of size five.
>
> We would like to emphasize the novelty and contribution of our paper. Using a path encoding to featurize neural architectures is a novel idea, which allows us to train a meta neural network to accurately predict the error of unseen neural architectures (Table 1, Figure 2, Figure 4). When combined with Bayesian optimization, this creates a novel NAS algorithm that achieves state-of-the-art performance on the NASBench and DARTS search spaces. We also carry out thorough and rigorous experiments, as we address every point on the NAS research checklist [Lindhaeur and Hutter, 2019].
>
> [1] Lakshminarayanan et al., Simple and Scalable Predictive Uncertainty Estimation using Deep Ensembles, 2017.
> [2] Beluch et al, The power of ensembles for active learning in image classification, 2018.
> [3] Choi et al, Ensemble of Deep Convolutional Neural Networks for Prognosis of Ischemic Stroke, 2017.

---

### Official Review · AnonReviewer2 · 2019-10-22
**Official Blind Review #2**

**Rating:** 3

**Review:**

This paper develop a path-based encoding scheme to featurize the neural architectures that are used to train the neural network model, and design a NAS algorithm that performs Bayesian optimization using a neural network model. The experiments show the priority of the proposed method.

In general, this paper is easy to follow, but the contribution is limited. The author did not give a clear explanation of why does this method work. There are several problems that exist in the paper:

1.	The paper introduced a path-based encoding scheme, which seems have nothing different from enumerating all possible paths. Any additional operations should be clarified in the paper.
2.	The method retains new architectures with high UCB value. However, the author did not prove that a higher UCB value leads to a better architecture. Eq.(1) trained several different networks to predict the accuracy. However, when using early stop stragegy, the intermediate accuracy is not convincing, and the new architecture selected based on UCB may not perform well when training with full epochs. If early stop is not used, there is no need  to predict the accuracy with different networks.
3.	In my opinion, Algorithm 1 is a simplified traditional Evolutionary Algorithm, which only have mutation operation and do not have selection and crossover operation, and has limited novelty.



**Experience Assessment:**

I have published in this field for several years.

**Review Assessment: Checking Correctness Of Derivations And Theory:**

I assessed the sensibility of the derivations and theory.

**Review Assessment: Checking Correctness Of Experiments:**

I assessed the sensibility of the experiments.

**Review Assessment: Thoroughness In Paper Reading:**

I made a quick assessment of this paper.

---

> ### Author Response · Authors · 2019-11-06
> **Reply to review #2**
>
> Thank you for your comments. We discuss them below.
>
> 1. Yes, the path-based encoding scheme is enumerating all possible paths, with no additional operations. As we show in Table 1 and Figures 2 and 4, using a path-based encoding is extremely effective for predicting the accuracy of neural architectures.
>
> 2. We do not optimize UCB to choose an architecture. We are optimizing the validation accuracy and only use UCB as an acquisition function to choose subsequent queries. This is the standard formulation for Bayesian optimization. In Bayesian optimization literature, UCB is a well-known acquisition function that has been used for efficient global optimization. In the NASBench experiments, there is no early stopping at all. Every architecture is trained for 108 epochs. In the DARTS experiments, every architecture is trained to 50 epochs (due to the large cost of training). Then at the very end of our algorithm, we take the architecture with the best validation accuracy and train it to 600 epochs. This method works well in practice.
>
> 3. Our method is completely different from an evolutionary algorithm. Our algorithm follows a standard Bayesian optimization procedure, which is a different class of strategies. In fact, we compared our algorithm to an evolutionary algorithm in Figure 2, and our algorithm performs much better. The only similarity is that we use a mutation function to optimize the acquisition function. Note that in the ablation study (Figure 3), we removed the mutation function (using random sampling instead), and our algorithm still significantly outperforms the evolutionary algorithm. There is no overlap between the two algorithms other than the mutation function. For example, our algorithm predicts the performance of unseen neural architectures, and there is no part of an evolutionary algorithm that makes predictions.

---

### Official Review · AnonReviewer1 · 2019-10-22
**Official Blind Review #1**

**Rating:** 3

**Review:**


The paper introduces a new encoding for cell structures to improve efficiency of neural architecture search methods.
As a second contribution the paper proposes Bayesian optimization with an ensemble of neural networks as probabilistic model for the objective function.
Both contributions combined show superior performance on the Nasbench101 benchmark as well as competitive performance on the DARTS search space.

While the paper identifies an important problem - encoding of architectures - I do not think the paper is ready for acceptance.

About the encodings:

First, using different encodings to enable better architecture search has been investigated by others before.
For example, Ying et al. also provided different encodings of the adjacency matrix for Nasbench101 besides the used binary encoding and it seems that different methods work well with different encodings.
Also in the work by Kandasamy et al. they presented an encoding for architecture, such that Bayesian optimization can be applied.

Second, the encoding described in the paper lacks some intuition.

 - How does enumerating all paths and encoding them as a binary vector convey more information than just using the adjacency matrix?

 - Lead isomorphic graphs, which for example occur in Nasbench101, to the same encoding?

 - It seems somewhat counter intuitive to use a large binary vector (more than 18000 dimensional vector for the DARTS space) as encoding for Bayesian optimization which is known to struggle with high dimensional input spaces.


About the Bayesian optimization strategy:

The proposed probabilistic model for Bayesian optimization seems straight forward and simple. Also here, previous work (Snoek et al, Springenberg et al., Perrone et al)  already proposed to use neural networks and ,in order to be more convincing, the paper should include a comparison to these methods.
Furthermore, the paper should clarify how the diversity in the neural network ensemble is enforced. Are the neural networks trained with different random initialization? How does it compare to the method proposed by Lakshminarayanan et al. which showed better performance for neural network ensembles based only on different random initialization?


Minor comments:


- In the Nasbench101 paper other Bayesian optimization strategies (e.g SMAC, BOHB, TPE) showed strong performance. The results would be more convincing if these methods are included in the comparison.

- Following the empirical protocol by Ying et al. the results would be easier to parse if the Figure 3  could report the log test regret on the y-axis. I am also missing a figure that shows the robustness of the method across independent runs.

- How are invalid architectures in the Nasbench101 (e.g architectures that violate the max edge constraint) treated in the experiments?

- I think the paper is missing the following references:

Simple and scalable predictive uncertainty estimation using deep ensembles
B Lakshminarayanan, A Pritzel, C Blundell
Advances in Neural Information Processing Systems, 6393-6395

Scalable hyperparameter transfer learning
V Perrone, R Jenatton, M Seeger, C Archambeau
Advances in Neural Information Processing Systems, 6845-6855



post rebuttal
------------------


I thank the authors for answering my questions regarding the path encoding and taking the time to improve the empirical evaluation of the paper. While I think that the paper has improved, I am afraid that the contributions of the paper are still not strong enough to reach the bar of acceptance because of the following reasons:

- The path encoding is an interesting approach and seems to improve upon just using the adjacency matrix directly, it doesn't scale and, hence, it remain somewhat unclear how valuable it is in practice.

- More importantly, I am not convinced that the proposed neural network model represents a sufficient contribution. After some discussion with the authors, they agree that existing BO methods based on neural networks could also be applied to this setting and even say that they may perform well with the path encoding. However, they are not include them in the comparison and only promise to add them for the final evaluation.  I am concerned that if it turns out that other methods perform as well or even better, it would dramatically lower the contribution of the paper.

**Experience Assessment:**

I have published in this field for several years.

**Review Assessment: Checking Correctness Of Derivations And Theory:**

N/A

**Review Assessment: Checking Correctness Of Experiments:**

I carefully checked the experiments.

**Review Assessment: Thoroughness In Paper Reading:**

N/A

---

> ### Author Response · Authors · 2019-11-06
> **Reply to review #1**
>
> Thank you for your helpful comments. We discuss the points below.
>
> Replying to comments on the encoding:
>
> First, we would like to point out the empirical effectiveness of our path encoding. We show in Table 1, Figure 2, Figure 3, and Figure 4 that the difference between the adjacency matrix encoding (from the NASBench paper) and the path encoding has a huge effect on the performance of both the meta neural network and the NAS algorithm. E.g., the NAS algorithm increases its efficiency by two orders of magnitude.
>
> The downside of the adjacency matrix encoding is that it gives an arbitrary ordering to the nodes, and then gives a binary feature for an edge between node i and node j, for all i, j. Then a list of the operations at each node must also be included in the encoding. This is a challenging data structure for a NAS algorithm to interpret. The other encoding used by Ying et al. is very similar to the adjacency matrix encoding, but the features of each edge are continuous. Our path encoding is quite different. We list all paths from the source to the sink in terms of the operations (Figure 1). There is no arbitrary node ordering, therefore, isomorphic graphs are automatically mapped to the same encoding. We will clarify these points in the final version of our paper.
>
> Also, Kandasamy et al. do not devise an encoding — they devise a distance metric between neural architectures, which is used in a GP model.
>
> We note that Bayesian optimization in general does not have a problem with high dimensions. Bayesian optimization is most commonly set up with a GP model, which struggles with high dimensions. We use a neural network model in Bayesian optimization, which is much more effective for high dimensional data.
>
>
> Replying to the comments on Bayesian optimization:
>
> Thank you for these suggestions. We cannot directly compare our algorithm to the three papers you cited (Snoek et al, Springenberg et al., Perrone et al), since they are methods for HPO rather than NAS. HPO typically involves optimizing a set of discrete or real-valued parameters, while the search spaces for NAS are complex DAG structures. Bayesian optimization has been used for HPO for years, but has only been applied to NAS last year. Still, we agree that we should discuss these papers in more detail (we cited Snoek et al, but we will cite and discuss the others too).
>
> Yes, in our neural network ensemble, the diversity is due to random initialization of the weights in the neural network and random order of the training data (same as Lakshminarayanan et al.) We will emphasize this point in the final version of our paper.
>
>
> Replying to minor comments:
>
> As we discussed at the start of page 4 and in the Experiments section, we did not compare against BOHB, HB, etc. because they are multi-fidelity NAS algorithms. Our algorithm is a black-box algorithm, and we compared to other black-box algorithms on NASBench. We mentioned a multi-fidelity version of our algorithm as exciting future work.
>
> We will include log test regret in the next iteration of the paper (although in many search spaces we cannot compute log test regret because the best architecture is unknown, e.g. on the DARTS search space).
>
> Please see Figure 3 for the error bars of our algorithm across different runs. We also discuss the error across runs in the last paragraph on page 7.
>
> We use the same random sampling method as in the original NASBench paper (rejection sampling). Our mutation function also uses rejection sampling (again, similar to the original NASBench paper). And just to clarify, we use the NASBench search space, so we reject cells with >9 edges.
>
> Thank you, we will include these references.
>
> We would also like to point out our code release and our slightly updated results on the DARTS search space. Our algorithm outperforms DARTS on average, and the best architecture achieves 2.57% test error on CIFAR-10. Therefore, we achieve state-of-the-art performance on multiple search spaces. Finally, we would like to emphasize that we carry out thorough and rigorous experiments, as we address every point on the NAS research checklist [Lindauer and Hutter, 2019].

---

> > ### Comment · AnonReviewer1 · 2019-11-12
> > **clarifications**
> >
> > Thank you for taking the time to answer my questions. However, there are still some points that need further clarifications:
> >
> > -  I don't follow the argument about the encodings: In the proposed encoding you still have to arrange the nodes somehow in order to find paths to the network? Also I still believe that in order to show that the proposed ordering helps to simplify the problem, it would be good if it is compare against other encodings that were proposed for Nasbench101 by Ying et al.
> >
> > -  I don't understand why the existing BO methods with neural networks cannot apply here? NAS without weight sharing boils down to the same thing as HPO, especially with the fixed sized discrete encoding proposed in this work (where apparently standard BO with GPs can be applied). Also, how is the architecture of the meta-network different to the commonly used architecture used by the aforementioned references?
> >
> > - You say that you compare against other blackbox algorithms but not against multi-fidelity  approaches, that is correct for HB and BOHB, but not for TPE and SMAC which are also BO methods and achieved strong performances on Nasbench101

---

> > > ### Author Response · Authors · 2019-11-14
> > > **clarification response**
> > >
> > > Thank you for your reply. These are all very good questions. We give our responses which we hope will clarify more.
> > >
> > > As you suggested, we ran BANANAS using the other encoding from the nasbench paper. It performed better than BANANAS with the adjacency matrix encoding (i.e., the main encoding from the nasbench paper), but still worse than BANANAS with our path encoding. This makes sense because both of the encodings from the nasbench paper use an arbitrary node indexing. We give the full results at the bottom.
> > >
> > > Let us explain our path encoding. Each feature of the path encoding corresponds with a directed path from the input to the output of an architecture cell (for example: input-->1x1-->3x3-->MP-->output). To encode an architecture, we simply check which of the possible paths it contains (i.e. which paths are present in the cell), and write this as a binary vector. Therefore, we do not need to arrange the nodes in any particular order for our encoding.
> > >
> > > Some more intuition: the features for the adjacency matrix are highly dependent on one another, e.g. an edge from the input to node 2 is useless if there is no path from node 2 to the output. Also, even if there is an edge from node 2 to the output, it matters a lot whether node 2 is a CONV or a POOL operation. In our path encoding, we don’t need to worry about either of these dependencies. In fact, at one point we computed the feature importance for prediction. We found that the path most correlated with validation accuracy is input—>CONV 3x3—>output (we think this is because architectures with only POOL operations have much worse validation accuracy than architectures with at least one CONV 3x3 operation). It would not make sense to run feature importance on the adjacency matrix encoding, since each feature is extremely dependent on the other features.
> > >
> > > Yes, you are correct that these HPO algorithms can be applied directly to NAS (using an encoding for architectures). We can compare our approach to the papers you cited in the final version of our paper. We predict that these three algorithms will not perform well with the adjacency matrix encoding, since they were designed to optimize over Euclidean space whereas NAS is optimizing over discrete DAG structures. Perhaps these algorithms may perform well if we use the path encoding (we view the path encoding as one of the main contributions of our paper).
> > >
> > > All three of the papers you cited use a feed forward neural network with three layers of size 50 with tanh activation, where the final layer computes predictions with uncertainty (using adaptive basis regression or Hamiltonian Monte Carlo). Apart from the last layer, these architectures are similar to our meta neural network architecture (ours is 5 layers of size 20 with ReLU activation). One of the biggest differences is that we use an ensemble approach instead of a Bayesian neural network (we preferred to use an ensemble for various practical reasons; see our reply to Anon Reviewer 3 for details/comparison between the two approaches).
> > >
> > > As you suggested, we ran TPE experiments. TPE struggled to perform better than random search (the nasbench paper also had the same findings). We will add SMAC experiments to the final paper as well (we are currently trying to run the automl/nas_benchmarks repo but are experiencing issues with SMAC3). We note that in our paper we did test regularized evolution (RE), and Figure 7 of the nasbench paper shows that RE outperforms SMAC, especially for <10^6 seconds (our setting).
> > >
> > > New results in anonymized link: https://drive.google.com/file/d/1vpKHMX975Ir2wmdtlVgnEzssWvKR3nO1/view?usp=sharing

---

> > > ### Author Response · Authors · 2019-11-15
> > > **Another quick clarification**
> > >
> > > One more quick clarification: the papers you cited that use neural networks in BO for HPO are significantly different than our work. The goal of Snoek et al. and of Springenberg et al. is to scale BO+GP by removing the matrix inversion step. They are focused on reducing the runtime of Bayesian optimization without affecting the accuracy-to-#-queries ratio. They use a neural network to emulate a Gaussian process.
> > >
> > > In our paper, we showed the path encoding + meta neural network achieves better accuracy than all other approaches. In particular, we compared to BO+GP (both with an adjacency-matrix distance metric and a path encoding distance metric), and our algorithm significantly outperformed them.

---

### Public Comment · ~Linnan_Wang1 · 2019-09-27
**Interesting work**

Hello there,

This is an interesting work, and I observed that the experiments are still within a very small scale on NASBench. To the best of my knowledge, Gaussian Process used in Bayesian Optimization suffers from the major scalability issue, e.g. it took a few hours to fit GP on 10^4 samples making it infeasible to tackle large scale search such as NAS. How do you tackle this problem? Thank you.

---

> ### Public Comment · ~Kaicheng_Yu1 · 2019-09-30
> **They provide experiments on DARTS space as well**
>
> After NASbench experiments, they also provided results on DARTS search space that is significantly larger. The result after 100 samples achieves 3.85 error rate. However, this result is not very convincing because DARTS itself got 3.75 (even we know it is via multiple trials, but still, better).

---

> > ### Author Response · Authors · 2019-10-01
> > **Our method has advantages over darts**
> >
> > Hi, thank you for your interest.
> > - Our method achieved error within 0.1% of DARTS - this is within the standard deviation across random seeds.
> > - Our method achieves good results in multiple search spaces, and beats all comparable algorithms on NASBench (where DARTS cannot be applied).
> > - More generally, there are other benefits of developing zeroth order (non-differentiable) NAS strategies, including that they allow for easy parallelism, integration with optimizing other hyperparameters, easy implementation, and optimization with respect to other non-differential objectives.

---

> ### Author Response · Authors · 2019-10-01
> **Our approach scales well**
>
> Hi, thank you for your interest in our paper.
>
> As explained in our paper, we use a neural network instead of a Gaussian process. Therefore, our method scales extremely well. For example, training a meta neural network on 10^4 NASBench architectures takes under 30 seconds, and for the DARTS search space it takes about 3 min (on a CPU).
>
> We believe 100-500 queries is an important setting for real-world NAS problems. For example, in the DARTS search space, it takes 15 GPU days to run any query-based NAS algorithm that trains 100 architectures for 50 epochs. Running a NAS algorithm that trains 1000 architectures would take 150 GPU days.

---

### Public Comment · ~Nikita_Klyuchnikov1 · 2019-10-16
**Neural networks used in the ensemble**

Dear authors,

I'm currently trying to reproduce the results reported in Table 1, so
could you disclose what neural networks you used in the ensemble (test error prediction model)?

---

> ### Author Response · Authors · 2019-10-23
> **Reproduce Table 1 using notebook**
>
> Hi, thank you for your interest in our paper. Please see the notebook meta_neuralnet.ipynb in the anonymous code in our comment above, which reproduces the results from Table 1 and Figure 4.

---

### Author Response · Authors · 2019-10-23
**Code release**

We release the code anonymously here: https://github.com/iclr474/bananas/

We also have slightly updated experimental results for the DARTS search space. In the original submission, we reported results without cutout and auxiliary tower. With cutout and auxiliary tower (which are the same hyperparameters used in the DARTS and RandomNAS papers), and a few minor tweaks to BANANAS, we achieve the following results:
- BANANAS: 2.64% mean, 2.57% best test error
- DARTS: 2.68% mean, 2.57% best test error
- ASHA: 3.08% mean, 2.98% best test error

---

### Author Response · Authors · 2019-11-15
**Updated paper**

Hello, thank you to all reviewers for helpful discussions. We updated our paper, with the main changes as follows:
- Updated experiments:
    - New results for the DARTS search space as in our last comment.
    - Comparison to TPE in the NASBench search space.
    - Comparison to BANANAS with the other encoding from the nasbench paper, in our ablation study.
- Discussion on more related papers, specifically papers on neural network ensembles, and HPO using BO with
  neural networks.
- More intuition of the path encoding.
- More clarifications.

---

### Author Response · Authors · 2020-01-30
**Post-review final update**

Hello all,
For the most up-to-date versions of our work, please refer to our arXiv paper here https://arxiv.org/abs/1910.11858
and our GitHub repository here: https://github.com/naszilla/bananas

For the sake of future readers of this OpenReview thread, we would also like to address the two reviewer concerns:

Concern (1): Our approach is similar to Springenberg et al. [1], Snoek et al. [2], and other works, and might not perform better than these approaches.

These papers are for hyperparameter optimization, but more importantly, the explicit goal of these papers is to improve the efficiency of Gaussian Process-based Bayesian optimization from cubic to linear time. In other words, they speed up the time it takes Bayesian optimization to choose the next neural network to try. In particular, the primary goal is NOT to develop a different type of prediction model in order to improve the performance of BO with respect to the number of iterations. For example, from the abstract of Springenberg et al. “We present a general approach ... staying as close to a truly Bayesian treatment as possible.”
Our work is different: we present new techniques which deviate significantly from Gaussian Process-based Bayesian optimization, and see a large performance boost with respect to the number of iterations (and our method scales linearly). Our new techniques are (i) the path encoding, and (ii) using a meta neural network to directly predict the performance of a neural architectures, foregoing the need for a distance function in Bayesian optimization. Both of these techniques are novel, to the best of our knowledge.
Because the goals of [1] and [2] stated above, we felt it suffices to compare BANANAS to standard GP Bayesian optimization (from the start, we included this comparison with the standard adjacency matrix as well as our path encoding). Since this was a major sticking point in the review process, we present new results directly comparing BANANAS to [2]. The result shows that [2] with the adjacency matrix encoding and path encoding both fall short of the performance obtained by BANANAS.

Concern (2): The path encoding does not scale with respect to the size of the neural architectures in the search space.

We have new experimental and theoretical results to show that simply truncating the path encoding scales well and works well in practice.
Look for an arXiv update very soon.

Thanks,
Colin, Willie, and Yash

[1]  Jasper Snoek, Oren Rippel, Kevin Swersky, Ryan Kiros, Nadathur Satish, Narayanan Sundaram, Mostofa Patwary, Mr Prabhat, and Ryan Adams. Scalable bayesian optimization using deep neural networks. In International conference on machine learning, pages 2171–2180, 2015.

[2] Jost Tobias Springenberg, Aaron Klein, Stefan Falkner, and Frank Hutter. Bayesian optimization with robust bayesian neural networks. In Advances in Neural Information Processing Systems, pages 4134–4142, 2016.

---

> ### Comment · AnonReviewer3 · 2020-02-11
> **question about BO with optimal transport**
>
> Dear authors,
>
> Your paper addresses an important ML problem of neural architecture search !
>
> I have read the paper and checked the code. I notice that the BO baseline is not a BO with optimal transport [1]. Your paper would be stronger if a comparison with [1] is demonstrated.
>
> Regards,
>
>
> [1] Kandasamy, K., Neiswanger, W., Schneider, J., Poczos, B., & Xing, E. P. (2018). Neural architecture search with bayesian optimisation and optimal transport. In Advances in Neural Information Processing Systems (pp. 2016-2025).
>
> Code: https://github.com/kirthevasank/nasbot

---

> > ### Author Response · Authors · 2020-02-20
> > **Now we compare to NASBOT**
> >
> > Hi reviewer 3,
> > Our new versions of the arxiv paper and github repository include a comparison to NASBOT. We could not use the original NASBOT repository because it is not set up for cell-based search spaces. Therefore, we designed an OTMANN-like distance that is compatible with cell-based search spaces. See our paper and repository for more details. (By the way, notice that Willie is an author of both NASBOT and BANANAS.)
> > Thanks,
> > Authors

---

### Decision · Program_Chairs · 2019-12-19

**Decision:**

Reject

**Comment:**

This paper uses Bayesian optimization with neural networks for neural architecture search.
One of the contributions is a path-based encoding that enumerates every possible path through a cell search space. This encoding is shown to be surprisingly powerful, but it will not scale to large cell-based search spaces or non-cell-based search spaces. The availability of code, as well as the careful attention to reproducibility is much appreciated and a factor in favor of the paper.

In the discussion, it surfaced that a comparison to existing Bayesian optimization approaches using neural networks would have been possible, while the authors initially did not think that this would be the case. The authors promised to include these comparisons in the final version, but, as was also discussed in the private discussion between reviewers and AC, this is problematic since it is not clear what these results will show. Therefore, the one reviewer who was debating about increasing their score did in the end not do so (but would be inclined to accept a future version with a clean and thorough comparison to baselines).

All reviewers stuck with their score of "weak reject", leaning to borderline. I read the paper myself and concur with this judgement. I recommend rejection of the current version, with an encouragement to submit to another venue after including a comparison to BO methods based on neural networks.